# RSP4J: An API for RDF Stream Processing

Riccardo Tommasini[1], Pieter Bonte[3], Femke Ongenae[3] and
Emanuele Della Valle[2]

1 University of Tartu, Data System Group, Estonia
`riccardo.tommasini@ut.ee`
2 Politecnico di Milano, DEIB, Milan, Italy
`emanuele.dellavalle@polimi.it`
3 Ghent University - imec, Ghent, Belgium
`firstname.lastname@ugent.be`

**Abstract.** The RDF Stream Processing (RSP) community has proposed several models and languages for continuously querying and reasoning over RDF streams over the last decade. They each have their semantics, making them hard to compare. The variety of approaches has fostered both empirical and theoretical research and led to the design of RSPQL, i.e., a unifying model for RSP. However, an RSP API for the development under RSPQL semantics was still missing. RSP community would benefit from an RSP API because it can foster comparable and reproducible research by providing programming abstractions based on RSPQL semantics. Moreover, it can encourage further development and in-use research. Finally, it can stimulate practical activities such as tutorials, lectures, and challenges, e.g., during the Stream Reasoning Workshop.

In this paper, we present RSP4J, a flexible API for the development of RSP engines and applications under RSPQL semantics. RSP4J offers all the necessary abstractions required for fast-prototyping of RSP engines under the proposed RSPQL semantics. Users can configure it to reproduce the variety of RSP engine behaviors in a comparable software environment. To promote systematic and comparative research, RSP4J is open-source, provides canonical citation, permanent web identifiers, and a comprehensive user guide for developers.

## 1 Introduction

The advent of the Internet of Things and social media has unveiled the streaming nature of information [9]. Data analysis should not only consider huge amounts of data from various complex domains, it should also be executed rapidly, before the data are no longer valuable or representative. Stream Reasoning (SR) is the research area that combines Stream Processing and Semantic Web technologies to make sense, in real-time, of vast, heterogeneous and noisy data streams [13].

Since 2008, the SR community's contributions include data models, query languages, and algorithms, and benchmarks for RDF Stream Processing (RSP).

As an extension of the Semantic Web stack, the value of RSP emerges in application domains where Data Variety and Data Velocity appear together [10], e.g., Smart Cities, e-Health and news analysis.

RSP approaches extend RDF and SPARQL to represent and process data streams. The community rapidly reached consensus around the use of RDF Streams as the data model. On the other hand, a variety of RSP languages emerged over time, e.g., C-SPARQL, CQELS-QL, $SPARQL_{stream}$, and Strider-QL. Such languages are extensions of SPARQL that support some form of continuous semantics. RSP languages are usually paired with working prototypes that helped proving the feasibility of the approach as well as studying its efficiency. Such variety of languages and systems enriches the state-of-the-art, but it may be hindering adoption and, thus, slow down the technological progress. Indeed, the diversity in the literature often opposes the identification of a clear winner, the establishment of best practices, and calls for comparative research.

Like other communities, e.g., OWL reasoning [20] and Big Data Systems [1], comparative research on and benchmarking of RSP engines is extremely hard. In fact, the semantics of different RSP languages do not completely overlap [12]. Moreover, the development of RSP engines, which are time-based systems, implies a number of design decisions that are often hidden in the code [7]. Such decisions, which fall into the notion of *execution semantics*[1], hamper the performance comparison, making it hard to reproduce the same behavior in two different systems and, thus, generalise the conclusions.

In summary, the lack of standardization and shared design principles are obstructing the growth of the communities. Indeed, as prototyping efforts remain isolated, the costs of development and maintenance of prototypes remain on the shoulder of individual researchers. Nevertheless, the problem did not remain unnoticed. The OWL reasoning community worked on shared APIs to standardise the evaluation of OWL reasoners [16], fostering a number of initiative like the OWL Reasoning Evaluation (ORE) challenges [20]. The Big Data Systems community witnessed the publication of a number of surveys and unification projects. In particular, Apache Beam is an attempt to uniform the APIs for stream and batch Big Data processing [17].

The RSP community is also working actively on solving this issue, focusing on (i) designing best practices [27, 24] (ii) disseminating the approaches [15], and (ii) developing benchmarks that take correctness and execution semantics into account [3, 18]. A recent important result is RSPQL [14], a reference model that unifies existing RSP dialects and the execution semantics of existing RSP engines. Although RSPQL is a first step towards a community standard, existing prototypes still do not follow shared design principles.

In line with the OWL APIs and Apache Beam initiative, *an API based on RSPQL would reduce the maintenance cost of existing engines, foster adoption of RSP engines, open new research opportunities in Stream Reasoning.*

In this paper, we present RSP4J, a configurable RSPQL API and engine, that builds on the lessons learned by developing the existing prototypes, and bringing

---

[1] also known as execution semantics

RSP research to the next level. We believe RSP4J can foster fast-prototyping, empirical and comparative research, as well as easing the dissemination of RSP via teaching. To this extent, RSP4J includes (i) all the necessary abstractions to develop RSP engines under the proposed RSPQL semantics and (ii) an implementation, i.e. YASPER, based on Apache Commons RDF[2], with the goal of showcasing the API's potential. Moreover, RSP4J can reproduce the variety of RSP engines in a comparable software environment.

In summary, the goals of RSP4J are (i) fostering the design and development of RSP engines under fixed RSPQL semantics, (ii) unifying the existing prototypes and their results, (iii) providing a framework for fair comparison of results and (iv) presenting a high-level API for easy adoption for RSP developers. RSP4J is open-source and is maintained on Github[3]. It has a canonical citation and permanent URL[4]. Moreover, it comes with an actively maintained documentation and a *Ready2GoPack* for increased availability to new members of the RSP community. RSP4J was already used in a number of tutorials and lectures, i.e. ISWC17, ICWE18, ESWC19, TheWebConf19, RW18/20.

The remainder of the paper is organized as follows: Section 2 discusses the potential impact of RSP4J in terms of use-cases, and afterwards presents the requirement analysis. Section 3 presents the background, concepts and definitions used throughout the rest of the paper, while Section 4 outlines architecture of RSP4J, its modules, and shows how it satisfies the requirements. In Section 5, we presents the related work. Finally, Section 6 concludes the paper and summarizes the most important contributions for RSP4J as a resource.

## 2    Impact: Use cases & Requirements for an RSPQL API

In this section, we discuss the potential impact of RSP4J as a resource. To this extent, we present different use cases that concern state-of-the-art prototypes for Stream Reasoning and RDF Stream Processing. We highlight the challenges that such use cases unveil, and we elicit a set of requirements for RSP4J in order to address such challenges. Table 1 summarizes the relationship between the challenges ($C_i$) and requirements ($R_j$).

### 2.1    Use cases

**Fast Prototyping.** In 2008, the first Stream Reasoner prototype came out [31]. Since then, the SR community has designed a number of working prototypes [5, 19, 8, 23], with the intent of proving the feasibility of the vision. E-health, smart cities, and financial transaction are examples of use cases where such prototypes were successfully used. Nevertheless, the effort of designing and engineering good prototypes is extremely high, and often their maintenance is unsustainable. In fact, prototypes are often designed with a minimal set of requirements and

---

[2] http://commons.apache.org/proper/commons-rdf/
[3] https://github.com/streamreasoning/rsp4j
[4] https://w3id.org/rsp4j

without shared design principles. In such scenarios (C1) adding new operators, (C2) new types of data sources to consume, or (C3) experimenting with new optimisations techniques requires huge manual efforts or is almost impossible.

**Comparable Research & Benchmarking.** Aside developing proof-of-concepts, the SR/RSP communities have focused a lot on Comparative Research (CR) [27, 24] and benchmarking [32, 22, 18, 3, 26]. CR studies the differences and similarities across SR/RSP approaches. Stream Reasoners and RSP engines can only be compared when they employ the same semantics. Thus, a fair comparison demands a deep theoretical comprehension of the approaches, a proper formulation of the task to solve, and an adequate experimentation environment [28]. Consequently, it is currently hard to (C4) reproduce the behavior of existing approaches in a comparable way. Moreover, experimentation is limited by (C5) the lack of parametric solutions, i.e. the configurability of the operators allowing to match engine behavior. On the other hand, research on benchmarking aims at pushing the technological progress by guaranteeing a fair assessment. While some of the challenges are shared with CR [27, 28], benchmarking is empirical research. To this extent, (C6) monitoring both the execution of continuous queries, as well as (C7) the engine behavior at run-time are of paramount importance. Unfortunately, not all the existing prototypes provide such entry points, and only black-box analysis is possible, e.g. it is impossible to measure the performance of each of the engine's internal operators.

**Dissemination.** Although SR research is at its infancy, a lot has been done on the teaching side. As prototypes and approaches reach maturity, several tutorials and lectures were delivered at major venues, including ICWE, ESWC, ISWC, RW, and TheWebConf [15, 10, 11]. These tutorials were often practical and aimed at engaging with their audience using simple yet meaningful applications. Nevertheless, existing prototypes were not designed for teaching purposes. Thus, they lack important features like the possibility to (C8) inspect the engine behaviors and (C9) they are not designed to ease the understanding at various levels of abstraction. Indeed, prototypes often (C10) neglect their full compliance to the underlying theoretical framework for practical reasons. Although this approach often benefits performance, it makes makes the learning curve more steep.

## 2.2   Requirements

Now we present the requirements that an RSPQL API should satisfy. We elicit the requirements from the challenges presented above. Although the requirements could be generalized for any RSP engine and Stream Reasoner, we restrict our focus to

| Req | C1 | C2 | C3 | C4 | C5 | C6 | C7 | C8 | C9 | C10 |
|-----|----|----|----|----|----|----|----|----|----|-----|
| $R_1$ | ✓ | ✓ | ✓ | | ✓ | | ✓ | | | |
| $R_2$ | | | | ✓ | | | | ✓ | | ✓ |
| $R_3$ | | | | | | | | ✓ | ✓ | ✓ |
| $R_4$ | | | ✓ | ✓ | ✓ | | | | | |
| $R_5$ | ✓ | ✓ | ✓ | | | ✓ | ✓ | | | |

Table 1: Challenges vs Requirements

Window-based RDF Stream Processing Engines, i.e., those covered by the RSPQL specification.

$R_1$   **Extensible Architecture**. An RSP API should allow the easy addition of data sources (C2) and operators (C1), and the design of optimization techniques (C3). Moreover, An RSP API should allow experimentation by allowing the addition of execution parameters (C5), and should ease the extension of engine capabilities (C7).

$R_2$   **Declarative Access**. An RSP API should be accessible in a declarative and configurable manner (C4). It should allow querying according to a formal semantics, e.g., RSPQL (C10), and should allow controlling the engine and the query lifecycles (C8).

$R_3$   **Programming Abstractions**. An RSP API should provide programming abstractions that allow interacting with the engine at various levels of abstractions (C9), abstractions that are based on a theoretical framework (C10), and that provide a blueprint to make sense of the engine behavior (C8).

$R_4$   **Experimentation**. An RSP API should be suitable for experimentation and, thus, should foster comparative research. To this extent, it should allow experimentation with optimizations techniques (C3), enabling to execute experiments using alternative configurations (C5). Last but not least, the reproducibility of state-of-the-art solutions should be a priority to enable replication studies (C4).

$R_5$   **Observability**. An RSP API should be observable by design, enabling the collection of metrics at different levels, i.e., stream level, operator level, query level (C6), and engine level (C7). Observability should be independent from architectural changes (C1 and C2), and ease study of optimizations (C3).

## 3   Background

In this section, we summarize the knowledge necessary to understand the main concepts of RSP4J. RSP4J is based on RSPQL, which in turn relies on the Continuous Query Language (CQL) [4] for its operation structure, SPARQL 1.1 semantics for RDF querying, and the SECRET model [7] for its operational semantics. Notably, we assume some knowledge on RDF and SPARQL semantics[5].

**Definition 1.** *A data stream $\mathcal{S}$ is an infinite sequence of tuples $\langle d_i, t_e, t_p \rangle$ where, $d_i$ is a data item, and $t_e/t_p$ are respectively the event time and the processing time timestamps. An **RDF Stream** is a stream where the data item $d_i$ is an RDF object and $t_e/t_p$ are timestamps indicating event time and processing time, respectively.*

In the literature, there are many definitions of data stream, with a general agreement on considering them as unbounded sequences of time-ordered data. Different notions of time are relevant for different applications. The most important ones are the time at which a data item reaches the data system (*processing*

---

[5] For a comprehensive analysis we suggest [21].

*time*), and the time at which a data item was produced (*event time*) [2]. In RSP, streams are represented as RDF objects, as stated by Definition 1  [14].

Operationally, stream processing requires a special class of queries that run under continuous semantics (vid. Definition 2). In practice, continuous queries consume one or more infinite inputs and produces an infinite output [25]. Arasu et al. [4] proposed a query model for processing relational streams based on three families of operators, as depicted in Figure 1. RSPQL extends these operators families to work on RDF Streams.

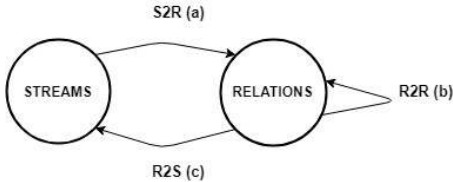

Fig. 1: The CQL query model, i.e., the S2R, R2R, and R2S operators.

**Definition 2.** *Under continuous semantics, the result of a query is the set of results that would be returned if the query were executed at every instant in time.*

*Stream-to-Relation* (S2R) (vid. Figure 1 a), i.e., is a family of operators that bridges the world of streams with the world of relational data processing. These operators chunk the streams into finite portions. A typical operator of this kind is a Time Window operator. In RSPQL, a time-based window operator is defined as in Definition 3.

**Definition 3.** *The time-based window operator $\mathbb{W}$ is a triple $(\alpha, \beta, t^0)$ that defines a series of windows of width ($\alpha$) and that slide of ($\beta$) starting at $t^0$.*

*Relation-to-Relation* (R2R) (Figure 1 b), i.e., is a family of operators that can be executed over the finite stream portions. In the context of RSPQL, R2R operators are SPARQL 1.1 operators evaluated under continuous semantics.

To clarify this intuition, Dell'Aglio et al. introduce the notion of a Time-Varying Graph and RSPQL dataset [14]. A **Time-Varying Graph** is the result of applying a window operator $\mathbb{W}$ to an RDF Stream $\mathcal{S}$ (vid. Definition 4), while the RSPQL dataset (SDS) is a an extension of the SPARQL dataset for continuous querying (vid. Definition 5).

**Definition 4.** *A **Time-Varying Graph** is a function that takes a time instant as input and produces as output an RDF Graph, which is called instantaneous.*
*Given a window operator $\mathbb{W}$ and an RDF Stream S, the Time-Varying Graph $TVG_{\mathbb{W},S}$ is defined where the $\mathbb{W}$ is defined.*

In practice, for any given time instant $t$, $\mathbb{W}$ identified a subportion of the RDF Stream S containing various RDF Graphs. The Time-Varying Graph function returns the union (*coalescing*) of all the RDF Graphs in the current window[6].

**Definition 5.** *An **RSPQL dataset** SDS extends the SPARQL dataset[7] as follows: an optional default graph $A_0$, n ($n \geq 0$) named Time-Varying Graphs, and m ($m \geq 0$) named sliding windows over k ($k \leq m$) data streams.*

---

[6] The current window identified by $\mathbb{W}$ with the oldest closing time instant at $t$
[7] https://www.w3.org/TR/rdf-sparql-query/#specifyingDataset

An RSPQL query is continuously evaluated against an SDS by an RSP engine. The evaluation of a RSPQL query outputs an instantaneous multiset of solution mappings for each evaluation time instant. The RSP engine's operational semantics determines the set $ET$ of evaluation time instants.

Finally, *Relation-to-Stream* (R2S) is a family of operators that returns to the world a set of infinite data from the finite ones, i.e., Figure 1 (c). RSPQL includes three R2S operators: (i) the **RStream** that emits the current solution mappings; (ii) the **IStream** that emits the difference between the current solution mappings and previous ones, and; (iii) the **DStream** that emits the difference between the previous solution mappings and the current ones.

When developing Stream Processing Engines to evaluate continuous queries there are a number of design decisions that might impact the query correctness. Such decisions, which are usually hidden in the query engine implementation, define the so called *operational semantics* (also known as execution semantics). Botan et al. [7], with their SECRET model, identified a set of four primitives that formalise the operational semantics of window-based stream processing engines. RSPQL incorporates these primitive and applied them on existing RSPQL engines: (i) *Scope* is a function that maps an event-time instant $t_e$ to the temporal interval where the computation occurs. (ii) *Content* is a function that maps a processing-time instant $t_p$ to the subset of stream elements included in the interval identified by the scope function. (iii) *Report* is a dimension that characterizes under which conditions the stream processors emit the window content. SECRET defines four reporting dimensions: (**CC**) *Content Change*: the engine reports when the content of the current window changes. (**WC**) *Window Close*: the engine reports when the current window closes. (**NC**) *Non-empty Content*: the engine reports when the the current window is not empty. (**P**) *Periodic* the engine reports periodically. (iv) *Tick* is a dimension that explains what triggers the report evaluations. Possible Ticks are: time-driven, tuple-driven, or batch-driven.

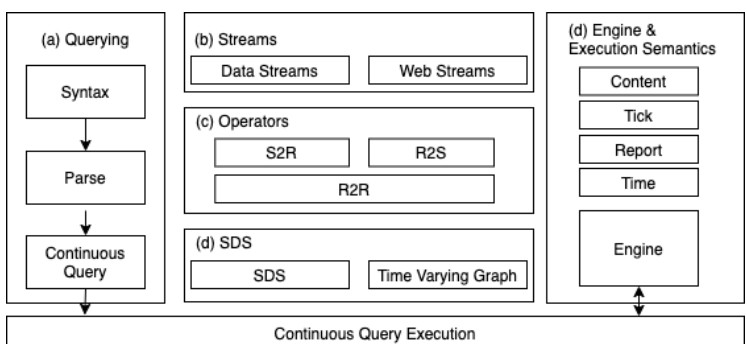

Fig. 2: RSP4J's Modules: (a) Querying (b) Streams, (c) Operators, (d) the SDS, and (e) Engine and Execution Semantics

## 4  RSP4J

In this section, we present RSP4J's s architecture, its components, and we show it satisfies the requirements (cf Section 2). Figure 2 shows RSP4J core modules, i.e., (a) Querying, (b) Streams, (c) Operators, (d) the SDS, and (e) the Engine with Execution Semantics. To provide concrete examples of RSP4J, we will use Yet Another Stream Processing Engine for RDF (YASPER). YASPER is a strawman proposal[8] designed for teaching purposes in the context of [15].

### 4.1  Querying

```
PREFIX : <http://example.org#>
REGISTER RSTREAM <output> AS
SELECT AVG(?v) as ?avgTemp
FROM NAMED WINDOW :w1 ON STREAM :stream1 [RANGE PT5S STEP PT2S]
WHERE {
    WINDOW :w1 { ?sensor :value ?v ; :measurement: ?m }
    FILTER (?m == 'temperature')
}
```

Listing 1.1: An example of RSPQL Query.

The query module contains the elements for writing RSPQL programs in a declarative way ($R_2$). The syntax is based on the proposal by the RSP community. At this stage of development, RSP4J accepts SELECT and CONSTRUCT queries written in RSPQL syntax (e.g. Listing 1.1)[9]. Although RSPQL [14] does not discuss how to handle multi-streams, RSP4J does, allowing its users to fully replicate the behavior of existing systems (cf $R_4$).

Moreover, RSP4J includes the `ContinuousQuery` interface that aims at making the syntax extensible (cf $R_1$). Indeed, RSP4J users can bypass the syntax module and programmatically define extensions in the query language.

### 4.2  Streams

The *Streams* module allows providing your own implementation of a data stream. It consists of two interfaces inspired by VoCaLS [30]: the `WebStream` and `WebDataStream`. `WebStream`, represents the stream as a Web resource, while `WebDataStream`, represents the stream as a data source. Figure 3 provides an overview of the relationships across these classes and interfaces. The `WebStream` does not include any particular logic. It is identi-

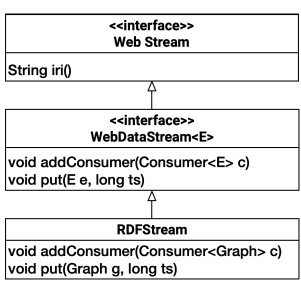

Fig. 3: Streams Interface.

fied by an HTTP URI so it can be de-referenced and then consumed through

---

[8] https://en.wikipedia.org/wiki/Straw_man_proposal
[9] The RSP W3C Community group has started working towards a common syntax and semantics for RSP (https://github.com/streamreasoning/RSP-QL).

an available endpoint [30]. Listings 1.11 shows the implementation of the `Web-DataStream` interface, which exposes two methods: `put` and `addConsumer`. The former allows injection of timestamped data items of type $E$ by producers; the latter connects the stream to interested consumers, e.g., window operators, or super-streams. The interface is generic, and it allows RSP4J's users to utilize multiple RDF Stream representation, i.e., either RDF Graphs or Triples, or even non-RDF Web Streams. A `WebDataStream` might also include some metadata relevant for the processing, i.e., links to ontologies, SHACL schemas, or alternative endpoints.

```
RDFStream implements WebDataStream<Graph> {
    protected List<Consumer<Graph>> cs = new ArrayList<>();
    @Override
    public void addConsumer(Consumer<Graph> consumer) {
        cs.add(consumer);}
    @Override
    public void put(Graph g,long ts) {
        cs.forEach(c -> c.notify(g, ts)); }
}
```

Listing 1.2: YASPER's WebDataStream implementation.

### 4.3 Operators

RSP4J core includes separate interfaces for all the RSPQL families of operators: *StreamToRelation*, *RelationToRelation*, and *RelationToStream*. These abstractions act both as lower level APIs for RSP4J's users (cf $R_3$) as well as a suitable entrypoint for extensions and optimizations (cf $R_1$). Moreover, each operator lifecycle could be monitored independently (cf $R_5$).

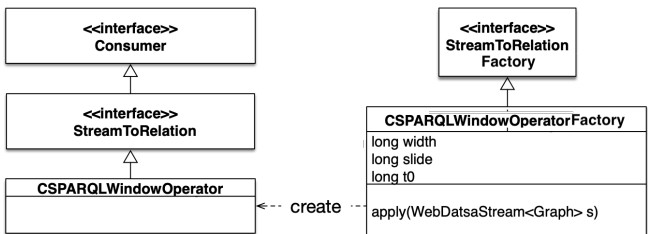

Fig. 4: UML class diagram for the S2R Package as used by C-SPARQL.

The **Stream To Relation** operator family bridges the world of RDF Streams to the world of finite RDF Data. RSPQL defines a *Time-Based Sliding Window* operators for processing RDF Streams. When applied to an RDF stream, RSPQL's S2R operator returns a function called Time-Varying Graph, that given a time instant $t$, materializes an Instantaneous (finite) RDF Graph.

```
public interface StreamToRelationOperator<I,O> extends Consumer {
    String iri();
    Report report();
    Tick tick();
    Content<I, O> materialise(long now);
    List<Content<I, O>> materialise(long now);
    TimeVarying<O> addConsumer(SDS<O> content);
    Content<I, O> compute(long t_e, Window w);
}
```

Listing 1.3: RSP4J S2R Operator Interface.

```
public interface StreamToRelationOperatorFactory<I,O> {
    String iri();
    TimeVarying<O> apply(WebDataStream stream);
    default boolean named() { return iri() != null;}
}
```

Listing 1.4: RSP4J S2R Operator Factory Interface.

To represent such behavior, RSP4J includes two interfaces, i.e., the **Stream-ToRelationFactory** interface and the **StreamToRelation** (S2R) operator. The former, exemplified in Listing 1.4, is used to instantiate the latter. It exposes the **apply** method that takes a generic **WebDataStream** as input, and returns a Time-Varying object **TimeVarying<O>**, decoupling the Type of the input stream content from the output Time-Varying Object. Listing 1.5 shows part of an implementation of the **StreamToRelationOperatorFactory** that instantiates C-SPARQL's *Time-Based Sliding Window*.

```
public class CSPARQLS2RFactory implements S2R<Graph,Graph> {
    private final long a, b, t0;
    private final ContinuousQueryExecution context;
    @Override
    public TimeVarying<Graph> apply(WebDataStream<Graph> s) {
     CSPARQLS2ROp op = new CSPARQLS2ROp(iri,a,b,scope,tick,report);
     s.addConsumer(op);
     return op.set(context);
    }
}
```

Listing 1.5: The CSPARQL's Time-Based Window Operator in YASPER.

The **StreamToRelation** operator is responsible for applying the windowing algorithm. In RSP4J, it is a special kind of **Consumer** that receives the data from the streams, cf Listing 1.3. Listing 1.5 shows that the factory instantiates a **CSPARQLS2ROp**, which is a **StreamToRelationOperator**, and registers it as stream consumer. Then, it obtains a **TimeVarying<Graph>** from the **ContinuousQueryExecution** context. We explain the details about the **TimeVarying<O>** when discussing the RSPQL Dataset $SDS$.

Figure 4 shows the UML class diagram of the S2R package. **CSPARQLWindowOperator** is a **StreamToRelation** operator, which creates a **TimeVarying<Graph>** if applied to a **WebDataStream<Graph>**.

In RSPQL, the **Relation To Relation** operator family corresponds to SPARQL 1.1 algebraic expressions evaluated over a given time instant. The evaluation of the Basic Graph Pattern produces a time-varying sequence of solution mappings, which can be consumed by SPARQL 1.1 operators.

```
public interface RelationToRelationOperator<T> {
    Collection<SolutionMapping<T>> eval(long ts);
}
```

Listing 1.6: RSP4J R2R Operator Interface.

Listings 1.6 shows the RSP4J interface that covers this functionality. Similarly to the S2R operators, the interface is generic to let the RSP4J's users decide the internal representation of the query solution, .e.g., the bindings.

The **RelationToStream** operator family allows going back from the world of Solution Mappings to RDF Streams. According to RSPQL the evaluation of an R2S operator takes as input a sequence of time-varying solution mappings. In RSP4J, we generalized this idea as shown in Listing 1.7, i.e., we allow the user to also provide the solution mapping incrementally as soon as they are produced.

```
public interface RelationToStreamOperator<T> {
  T eval(SolutionMapping<T> sm, long ts);
  default Collection<T>
  eval(Collection<SolutionMapping<T>> s, long t){
   return s.stream().map(sm -> eval(sm, t)).collect(Collectors.toList());
    }
}
```

Listing 1.7: RSP4J R2S Operator Interface.

### 4.4 SDS and Time-Varying Graphs

Like in SPARQL, the query specification and the SDS creation are closely related. An RSPQL dataset SDS is an extension of the SPARQL dataset to support the continuous semantics. As indicated in Section 3, the SDS is time-dependent as it contains Time-Varying Graphs. RSP4J includes both the abstractions, i.e., the SDS and the TimeVarying Graphs (cf $R_3$).

Listing 1.8 shows RSP4J's `SDS` interface. The generic parameter is inherited by the generic nature of RSP4J's Time-Varying objects. The `consolidate` method consolidates the SDS content by recursively consolidating every Time-Varying Object it contains.

```
public interface SDS<E> {
    default void consolidate(long t) {
        asTimeVaryingEs().forEach(tvg -> tvg.consolidate(ts));
    } ...
}
```

Listing 1.8: RSP4J's SDS Interface.

Listing 1.9 shows a Time-Varying Graph that is the result of the application of the Window Operator to an RDF Streams. The method `materialize` consolidates the content at a given time instant *ts*. To this extent, it exploits the

`StreamToRelationOperator` interfaces, freezing and polling the active window content. The *coalesce* method ensure only one graph, among those selected during the windowing operation, is returned. According to RSPQL such graph corresponds to the union of the RDF graphs in the window.

```java
public class TimeVaryingGraph implements TimeVarying<Graph> {
    private IRI name;
    private StreamToRelationOperator<Graph, Graph> op;
    @Override
    public void materialize(long ts) {
        graph = op.getContent(ts).coalesce(); }...
}
```

Listing 1.9: YASPER's Time Varying Graph Implementation.

As time progresses, the $SDS$ is reactively consolidated into a set of (named) Instantaneous Graphs[10] at the time $t$ at which a Time-Varying Graph is updated.

Therefore, RSP4J includes the `SDSManager` and `SDSConfiguration` interfaces. The former controls the creations, detection, and the interactions with the `SDS`; ideally this represents a starting point for federated query answering and/or multi-query optimisation. The latter makes the execution parametric e.g., for enabling different approaches for window management, or alternative output serializations, e.g., JSON-LD or Turtle.

### 4.5   Engine, Query Execution, and Execution Semantics

This module includes the abstractions to control and monitor the engine and the query lifecycle (cf $R_4$ and $R_5$). Moreover, we explain RSP4J's parametric execution semantics ($R_4$).

The **Engine** interface allows controlling RSP4J's capabilities, e.g., query registration and cancellation. It is based on the VoCaLS service feature idea [30]. Each engine can implement different interfaces, each of which correspond to a particular feature. By querying the implemented interfaces it is possible to list all the features exposed by the engine of choice, e.g., stream registration, RSPQL support, or formatting the results in JSON-LD format.

RSP4J can reproduce the **execution semantics** of common RSP engines by configuring SECRET's primitives: `Tick` is represented as an enumeration, i.e., tuple-based, time-based, and batch. `Scope` is a parameter accessible through the `Time` interface. `Time` controls the time progress w.r.t. the stream consumption. It is initialized with the system initial timestamp at configuration time. It keeps track of the evaluation timestamps $ET$, and exposes the time-progress to the user both for event-time and processing-time[11]. `Report` is represented as a collection

---

[10] Slowly evolving RDF graph are represented as a (named) Time-Varying Graph too.

[11] RSPQL determines the evaluation time instant set $ET$ wrt the reporting policy and the input data. Instead, RSP4J serves time as it receives data, i.e., by consuming the streams. Thus, RSP4J's $ET$ is built progressively. While the RSPQL's $ET$ is deterministic, RSP4J $ET$ might not be deterministic in case of distributed computations.

of `ReportingStrategies`. RSP4J core includes RSPQL's reporting policies, i.e., On-Content-Change, Non-Empty-Content, Periodic, and On-Window-Close. Last but not least, the `Content` interface represents the data items in the active window. It is generic and exposes the *coalesce* allows alternative implementations of the Time-Varying Graph functions.

```
public interface Time {
    long getScope();
    long getAppTime();
    void setAppTime(long now);
    ET getEvaluationTimeInstants();
    void addEvaluationTimeInstants(TimeInstant i);
    default long getSystemTime() {
        return System.currentTimeMillis();}
}
```

Listing 1.10: RSP4J's Time Interface.

The **ContinuousQueryExecution** interface represents the ever-lasting computation required by continuous queries. It allows monitoring and controlling the query life-cycle. Moreover, in order to make observable ($R_5$) the SDS and the operators involved in querying, the interface includes getters.

```
public interface ContinuousQueryExecution<I, E1, E2> {
    <O> WebDataStream<O> outstream();
    ContinuousQuery getContinuousQuery();
    SDS<E1> sds();
    StreamToRelationOperator<I, E1>[] getS2R();
    RelationToRelationOperator<E2> getR2R();
    RelationToStreamOperator<E2> getR2S();...
}
```

Listing 1.11: An example of RSP-QL Query.

## 5   Related Work

In this section, we present the work related to RSP4J. We present the most popular RSP engines, and how they differ in terms of RSPQL semantics, complicating fair comparison.

The C-SPARQL Engine [5] is an RSP engine that adopts a black box approach by pipelining a DSMS system with a SPARQL enige. The DSMS is used to execute the S2R operators and the execution semantics, while the SPARQL engine performs the evaluation of the queries implemented as the R2R operator. C-SPARQL supports the Window Close and Non-empty content reporting policies while employing RStreams as R2S operators.

The CQELS Engine [19] takes a white box approach, such that it has access to all the available operators, allowing it to optimize query evaluation. Compared to C-SPARQL, it supports the Content Change reporting strategy. Furthermore, CQELS supports the IStream R2S operator instead of the RStream.

Morph$_{stream}$ [8] focuses on querying virtual RDF streams with SPARQL$_{stream}$. Thus, compared to C-SPARQL and CQELS, it uses Ontology Based Data Access

to virtually map raw data to RDF data. Similar to C-SPARQL it supports the Window Close and Non-empty content reporting policies. Morph$_{stream}$ is the only engine that supports all R2S operators.

Strider [23] is a hybrid adaptive distributed RSP engine that optimizes the logical query plan according to the state of the data streams. It is built upon Spark Streaming and borrows most of its operators directly from Spark. Strider translates Strider-SQL queries to Spark Streaming's internal operators. It inherits the Window Close reporting policy from Spark Streaming, and supports the RStream as R2S operator.

In addition to the rigid yet explicit characteristics that each engine has, they also have inherent subtle differences. For example, none of them allows to define the starting timestamp $t^0$ as part of the time-based sliding window operators definition. This means that the starting time is supplied by the engine itself and in case of processing time engines are bound to produce different results. Differences like the $t^0$ make impossible to correctly compare the results produced by the various engines. RSP4J allows to customize the inner wiring of the engines, in order to align their semantics and allowing them to produce comparable results.

## 6    Conclusion, Discussion, and Roadmap

In this paper, we presented RSP4J, a flexible API for RSP development, adhering to the semantics of RSPQL, and YASPER, i.e., a strawman RSP engine implementation designed for teaching purposes in the context of RW [15].

RSP4J aims at solving three use case: fast prototyping, benchmarking comparative research, and dissemination via teaching. Thus, we designed it to fulfill, i.e., a set of requirements: (I) an extensible architecture, (II) declarative access through a uniform query language according to the RSPQL semantics; (III) the necessary programming abstractions; (IV) enable experimentation and fair comparison and (V) observability by design. In Section 4, we explained how each RSP4J module solves a subset of requirements. Differently than the state-of-the-art RSP prototypes which only solve requirement $R_2$ by providing a declarative access, RSP4J fulfills all the set requirements (cf Section 4). Moreover, two RSP engines already bind to RSP4J: (i) YASPER[12], which is a strawman implementation based on Apache RDF Commons[2], and C-SPARQL 2.0[13] a new version of the C-SPARQL engine [5].
**Roadmap.** RSP4J's future work includes a number of initiatives. We plan to bind even more engines, i.e., Morph$_{stream}$ and CQELS and run a reproducibility challenge in the context of the upcoming stream reasoning workshop. In the mid term, we would like to abstract RSP4J specification and provide access in other languages, e.g., Python. In the long-term, we would like to include abstraction to control the stream publication lifecycle [29]. Moreover, we would like to investigate how to combine RSP4J with other stream reasoning framework [6].

---

[12] `https://github.com/streamreasoning/rsp4j/tree/master/yasper`
[13] `https://github.com/streamreasoning/csparql2`

**Acknowledgments.** Dr. Tommasini acknowledges support from the European Social Fund via IT Academy program, and from the European Regional Development Funds via the Mobilitas Plus programme (grant MOBTT75). Moreover, the authors would like to acknowledge the support of Robin Keskisärkkä and Daniele Dell'Aglio in earlier versions of this work.

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
