# OpenReview forum: "RSP4J: An API for RDF Stream Processing"
_eswc-conferences.org/ESWC/2021/Conference/Resources_Track — ESWC 2021 Resources_

### Official Review · AnonReviewer2 · 2021-01-08
**The paper presents an API for interacting with RSP engines. It claims that it abstract over the semantics of the underlying engine to provide common RSPQL functionality.**

**Rating:** 2
**Confidence:** 4

**Review:**

The paper presents a API abstraction for working with RSP engines. The work is well motivated and presents the challenges that the API is to address. To date, the API has only been shown to work with YASPER and C-SPARQL. It is claimed that it should be able to abstract over the operational semantics of other engines such as CQELS, but this has not yet been demonstrated.

The source code is made available and has a suitable license on the public repository. The paper claims that there are rich documentation and Ready2Go pack, but it is not clear to me where these are located. All I could find was the GitHub wiki pages which did not allow me to get going with the resource. Perhaps links to the conference tutorials that have been given could be added to the repository's README file. The source code did compile with maven.

## Minor points:

- Link to source code could be given in the abstract
- Query listings should include a statement of the intent of the query as a comment at the start of the query
- Cross-reference for Listing 1.2 at top of p9 is wrong
- Minor typographical errors throughout which can be resolved with a proofread

## Post rebuttal

I look forward to reading the authors continued work in this area. My concerns should not prevent this paper being published as a resource paper.

**Anonymity:**

No, I would like my review to be deanonymized.

**Strong Points:**

- Link to public source code repository with clear license statement
- Used for community tutorials over a number of years
- Promising framework to support the RSP community
- Comprehensive review of the field
- Clear statement of challenges

**Subreviewer:**

I submitted this review.

**Weak Points:**

- Only shown to work with the YASPER engine so far
- Missing documentation to get going with the API
- No comparison experiments performed over different engines
- Over reliance on the reader remembering all the acronyms introduced. There is no need to abbreviate everything.

---

> ### Author Rebuttal · Authors · 2021-01-28
>
> We would like to thank the reviewer for their valuable insights. Regarding the specific comments:
>
> - Limited engine support:
> At this point, we support YASPER, i.e. strawman implementation based on Apache RDF Commons, and C-SPARQL 2.0 a renewed version of the C-SPARQL engine. We are currently working on supporting other engines such as CQELS, Morphstream and Strider.
>
> - Missing get going documentation:
> We are actively extending the documentation to extend it and include further material that would foster the reuse.
>
> - No comparison experiments performed over different engines:
> We agree that this would be very valuable. We are currently working on supporting other engines. Once this task is completed, we will work towards a reproducibility study of existing RSP engines.
>
> - Over-reliance on the reader remembering all the acronyms introduced
> In hindsight, we agree that many acronyms were introduced through the text. We will revise the manuscript such that only the most important and recurring terms are abbreviated.

---

> > ### Comment · AnonReviewer2 · 2021-02-02
> > **Post rebuttal comment**
> >
> > I look forward to reading the authors continued work in this area. My concerns should not prevent this paper being published as a resource paper.

---

### Official Review · AnonReviewer5 · 2021-01-14
**A Java API pour RDF stream processing**

**Rating:** 2
**Confidence:** 3

**Review:**

The paper proposes a Java API for RDF stream processing systems with the objective to standardise them and allow better benchmarking. It is based on RSPQL, a W3C proposal for an RDF Stream Processing Query Language.
The proposal is soundly based, and relates to the most famous stream processing engines of the literature.

I acknowledge I read and took into account the authors response

**Anonymity:**

Yes, I would like my review to remain anonymous.

**Strong Points:**

The objective of RSP benchmarking is relevant and timely ; providing an API for such an objective is a first step.
The paper is very well written.
The associated Github resource provides also Yasper, an implementation of this API.

**Subreviewer:**

I submitted this review.

**Weak Points:**

We wait for the associated benchmark!

---

> ### Author Rebuttal · Authors · 2021-01-28
>
> We would like to thank the reviewer for the positive feedback. We can already reveal that the benchmark will be published soon.

---

> > ### Comment · AnonReviewer5 · 2021-02-02
> > **I read the rebuttal**
> >
> > I acknowledge I read and took into account the response

---

### Official Review · AnonReviewer3 · 2021-01-15
**RSP4J: An API for RDF Stream Processing**

**Rating:** 2
**Confidence:** 4

**Review:**

The submission presents an API for RDF Stream Processing, which is clearly in the scope of the conference track. The API covers the RSPQL model and should make engine implementations easier to compare and use. Two implementations, C-SPARQL and YASPER, of the API are mentioned (the latter is a toy prototype for teaching purposes). Presentation is in general quite clear (modulo the typos listed below).

Operational semantics is a standard term in programming languages (cf. denotational semantics), and the way the authors "reuse" it can be quite misleading. Botan et al. [7], for example, refer to it as execution semantics, which does not have any other connotations and adequately represents the idea.


too many "foster" in the abstract

"populated" is not quite the right word (p 1)

application domains (no s, p 1)

shoulder*s* of individual researchers (p 2)

there is no verb "uniform" (to unify?, p 2)

OWL API (singular, p 2)

adaption -> adoption (p 3)

the URLs in footnotes are typeset in different fonts - is there any reason for that? (p 3)

Education is an unexpected paragraph heading (p 4). Dissemination?

necessary knowledge -> knowledge necessary (p 5)

use \langle and \rangle for tuples, not < and > (p 5)

the meaning of non-decreasing when applied to a single number (which t_e is) is unclear; "the data item d_i" also refers to a single element (p 5)

the abbreviation "cf." is misused here and in many other occurrences (it does *not* mean "see")

"Arasu et al." requires a citation after it (p 6)

in Definition 3, the second "that" seems to be a typo (p 6)

in Definition 4, a*n* RDF graph (p 6)

in Definition 5, there are too many colons (p 6)

a*n* RSPQL query (p 7)

((i) looks strange (p 7)

the nested list inside (iii) is a bit unexpected and difficult to get through; also, the use of both numerical and symbolic labels is redundant - the items could be (CC), (WC), (NC) and (P) (p 7)

is *a* straw-man proposal (p 8)

"does support to query them" is broken (p 8)

"allows to provide", "allows to go back", etc. require an object (e.g., "us") (pp 8, 11, 14)

a*n* HTTP (p 8)

RSPl-QL -> RSPQL (p 9)

typo in Listing 1.4 (p 9)

the the UML schemas -> the UML class diagram (p 10)

.e.g., -> e.g., (p 11)

enige -> engine (p 13)

*a* straw-man (p 14)

future works -> future work (p 14)

the meaning "To these extents, it is designed to fulfill, i.e.," is very unclear (p 14)

"renewed version" is an unexpected expression (p 14)


**Anonymity:**

Yes, I would like my review to remain anonymous.

**Strong Points:**

The API that would unify various RDF stream processing engines was clearly missing, so the submission is very topical (if not overdue).

The submission is reasonably well-written.


==
I acknowledge the rebuttal.

**Subreviewer:**

I submitted this review.

**Weak Points:**

A very minor weakness is the slightly misused term "operational semantics".

---

> ### Author Rebuttal · Authors · 2021-01-28
>
> We thank the reviewer for the positive comments. We regret the abundance of remaining typographical errors in the text. We thank the reviewer for pointing out many of them. We will revise the document in detail.
>
> Regarding the use of `the term "Operational semantics" in place of ‘execution semantics’. We agree with the reviewer on the potential source of ambiguity from the former, and we agree that the latter is more adequate to the context. We will update the terminology throughout the paper.

---

### Official Review · AnonReviewer4 · 2021-01-15
**The paper presents an API for stream reasoning enabling the simple and straightforward implementation of various stream reasoning applications**

**Rating:** 2
**Confidence:** 5

**Review:**

RSP community was working for more than a decade on the development of various approaches to stream reasoning. However, these developments were isolated from each other sharing only a common data format - RDF. The paper presents an API that is designed to allow for development, maintenance, integration, and comparison of different stream reasoning applications. This API is not only of interest to the scientific community, for instance, by simplifying the reuse of different components and their comparison, but also should enable its usage by software developers, who would like to use the results of the research in their applications. In addition, such an API can be used as a unified interface to the existing stream processing systems, triple stores, or other standard interfaces used in practical applications, such as the Internet of Things or Industry 4.0

**Anonymity:**

Yes, I would like my review to remain anonymous.

**Strong Points:**

The authors provide a comprehensive analysis of challenges end requirements for the API which justifies their choices. The paper is well-structured and is easy to read. Overall, I find the work quite important for the stream reasoning community and recommend this paper to be accepted to the conference.

**Subreviewer:**

I submitted this review.

---

> ### Author Rebuttal · Authors · 2021-01-28
>
> We like to thank the reviewer for the very positive comment!

---

### Official Review · AnonReviewer1 · 2021-01-16
**RSP4J: An API for RDF Stream Processing**

**Rating:** 3
**Confidence:** 5

**Review:**

This submission presented RSP4J, an JAVA API for the the RDF Stream Processing (RSP) community. It shows that the API has been adopted in many cases and are used in a number of events. This shows that the API has been widely accepted in the RSP community. This is similar to the roles of RDF4J for the RDF community and OWLAPI for OWL community.  Therefore, RSP4J is a great contribution.

The paper has clearly presented the design of the API, and the formal semantics. Many examples are provided to explain all the important aspects. It is also encouraging to see the two implementations of RSP4J in the systems YASPER and C-SPARQL 2.0.

**Anonymity:**

Yes, I would like my review to remain anonymous.

**Strong Points:**

- Clear explanation of the project
- Strong adoption of the API
- Community around the RSP4J

**Subreviewer:**

I submitted this review.

**Weak Points:**

- There are some small formatting issues. The URLs in footnotes do not have the same appearance.

---

> ### Author Rebuttal · Authors · 2021-01-28
>
> We like to thank the reviewer for the positive comments and for noticing the formatting issues. We will go through the paper in great detail to fix these mistakes.

---

### Decision · Program_Chairs · 2021-02-23

**Decision:**

Accept

**Comment:**

The resource is a standard set of programming interfaces useable for various RDF stream processing engines from the literature.

The reviewers are confident that this is an important and FAIR contribution to the stream reasoning community, and unanimously recommend this paper for acceptance to the resource track.